## Research Article

# Mechanistic aspects of maltotriose-conjugate translocation to the Gram-negative bacteria cytoplasm

Estelle Dumont[1,*], Julia Vergalli[1,*], Jelena Pajovic[2,3], Satya P Bhamidimarri[4], Koldo Morante[4], Jiajun Wang[4], Dmitrijs Lubriks[5], Edgars Suna[5], Robert A Stavenger[6], Mathias Winterhalter[4], Matthieu Réfrégiers[2], Jean-Marie Pagès[1]

**Small molecule accumulation in Gram-negative bacteria is a key challenge to discover novel antibiotics, because of their two membranes and efflux pumps expelling toxic molecules. An approach to overcome this challenge is to hijack uptake pathways so that bacterial transporters shuttle the antibiotic to the cytoplasm. Here, we have characterized maltodextrin–fluorophore conjugates that can pass through both the outer and inner membranes mediated by components of the *Escherichia coli* maltose regulon. Single-channel electrophysiology recording demonstrated that the compounds permeate across the LamB channel leading to accumulation in the periplasm. We have also demonstrated that a maltotriose conjugate distributes into both the periplasm and cytoplasm. In the cytoplasm, the molecule activates the maltose regulon and triggers the expression of maltose binding protein in the periplasmic space indicating that the complete maltose entry pathway is induced. This maltotriose conjugate can (i) reach the periplasmic and cytoplasmic compartments to significant internal concentrations and (ii) auto-induce its own entry pathway *via* the activation of the maltose regulon, representing an interesting prototype to deliver molecules to the cytoplasm of Gram-negative bacteria.**

## Introduction

The rise of multidrug resistance in pathogens is a serious and a growing worldwide threat (1) (Centers for Disease Control and Prevention, https://www.cdc.gov/drugresistance/biggest_threats.html; European Centre for Disease Prevention and Control, https://ecdc.europa.eu/sites/portal/files/documents/AMR-surveillance-Europe-2016;

World Health Organization, http://www.who.int/medicines/areas/rational_use/antibacterial_agents_clinical_development/en/). A significant scientific challenge in discovering new, effective antibacterial agents is penetration of active molecules into bacteria to elicit activity (1, 2). This aspect is particularly difficult for Gram-negative bacteria which have two membranes, the outer (OM) and the inner membranes, providing a barrier to the intracellular accumulation of molecules (3, 4). In addition, efflux pumps can efficiently expel compounds (5, 6, 7, 8, 9, 10, 11), which can contribute to the intrinsic resistance of Gram-negative bacteria to many antibacterials. The antibiotic concentration at the site of action must exceed a level where it can bind and exert its antibacterial activity, and influx of the antibiotic, e.g., permeation across the Gram-negative envelope, is essential to access its target (8, 9, 11, 12, 13, 14).

With the spread of multidrug resistant bacteria, several ways have been explored to bypass membrane-associated mechanisms of resistance including: (a) improved understanding of the porin pathways for penetration, (b) studying combinations of antibiotics with OM permeabilizers, (c) nanoparticle complexes to improve penetration, and (d) use of "Trojan Horse" approaches to hijack active uptake pathways and capitalize on existing bacterial receptors-transporters (15, 16, 17, 18, 19, 20, 21, 22). Multiple "Trojan Horse" approaches have been reported, including iron uptake via siderophore-conjugates and Opp-dependent peptide uptake pathways, but these systems are highly complex involving many receptors, and we are not aware of any that have been characterized by measuring the increase in intracellular accumulation. Another attractive strategy is to use the maltodextrin pathway, which is responsible for the internalization and degradation of maltodextrins in the cytoplasmic space of Gram-negative bacteria (23, 24). This system comprises the maltoporin (LamB) that provides an efficient pathway for maltose penetration through the OM, the

[1]Aix Marseille Univ, Institut National de la Santé et de la Recherche Médicale, Service de Santé des Armées, Institut de Recherche Biomédicale des Armées, Membranes et Cibles Thérapeutiques, Marseille, France   [2]DISCO Beamline, Synchrotron Soleil, Saint-Aubin, France   [3]University of Belgrade, Faculty of Physics, Belgrade, Serbia   [4]Department of Life Sciences and Chemistry, Jacobs University Bremen, Bremen, Germany   [5]Latvian Institute of Organic Synthesis, Riga, Latvia   [6]Antibacterial Discovery Performance Unit, Infectious Diseases Discovery, GlaxoSmithKline, Collegeville, PA, USA

Correspondence: jean-marie.pages@univ-amu.fr
Satya P. Bhamidimarri's present address is Institute for Cell and Molecular Biosciences, The Medical School, Newcastle University, Newcastle, UK
Jiajun Wang 's present address is Laboratory for Advanced Materials, School of Chemistry and Molecular Engineering, East China University of Science and Technology, Shanghai, China
*Estelle Dumont and Julia Vergalli contributed equally to this work

maltose binding protein (MalE) located in periplasm, maltodextrin transporters in the inner membrane (MalF, MalG, MalK) and various cytoplasmic degradative enzymes; these components are under the control of regulators belonging to maltose regulon (23). The internalization of a maltohexaose fluorophore (25), the uptake of a thiomaltose–trimethoprim conjugate (26), and the antibacterial activity of a radezolid analog (patent WO 2016/044846) have been recently reported. Interestingly, labeled-maltodextrins have also been used to study the localization/diagnosis of bacterial infections in mouse models of infection (27, 28).

To molecularly dissect the potential of the maltodextrin transport system for drug transport, a collaboration between IMI TRANSLOCATION and IMI ENABLE (www.imi.europa.eu) was established to study the uptake of maltodextrin conjugates. We report our results on both a maltohexaose–perylene conjugate (analog 1 in reference 25) and a shorter maltotriose–perylene conjugate, wherein the perylene moiety offers a handle for conjugate detection via fluorescence while mimicking a potential antibacterial "payload."

To provide molecular insight into the transit of the maltodextrin conjugates, uptake was studied in purified LamB porin reconstituted in lipid bilayers that demonstrated that the conjugates were able to pass through the LamB channel. By using spectrofluorimetry and microspectrofluorimetry (29, 30, 31, 32), we further demonstrated a LamB-dependent uptake of one conjugate into the periplasmic and cytoplasmic space of the bacterial cells. In addition, we have shown that the maltotriose perylene conjugate can promote the expression of maltose-binding protein belonging to the maltose regulon, consistent with the uptake of the maltotriose conjugate.

## Results

### Accumulation of Cpd-1 and Cpd-2 is dependent on LamB expression

To study the ability of Cpd-1 (maltotriose–perylene conjugate) and Cpd-2 (maltohexaose–perylene conjugate) (see Fig 1 and the

Materials and Methods section of the Supplementary Information for synthesis) to translocate across the *Escherichia coli* envelope, a series of well-defined isogenic strains were used, based on the parental strain (RAM1292) which contains an intact maltose operon (33). A *lamB* knock-out strain was obtained (RAM2806), as well as the *lamB* knock-out strain containing an empty pBAD24 plasmid (RAM2807), and the *lamB* knock-out strain with the pBAD24 plasmid coding for *lamB* under the control of an arabinose inducible promoter (RAM2808) (Fig S1). The strains did not differ significantly in OmpC or OmpF content, whereas LamB expression was shown to be absent under normal growth conditions. As expected, LamB was detected in RAM2808, but not in RAM2807, after induction with arabinose (Fig S1).

In addition, in the strains RAM2808 and RAM1292, MalE protein was highly expressed in the presence of exogenous maltose consistent with induction of the maltose operon (Fig 2A). When bacterial cells are grown with high glucose concentrations, the catabolic repression blocks the transcription of *mal* genes. In contrast, under limited glucose concentration, the maltose regulon is expressed at elevated levels (23) (Figs S2, S3).

To study the implication of LamB and MalE in the translocation of conjugates, strains RAM1292 and RAM2808 were grown under the following conditions: RAM1292 cells were grown under conditions of maltose operon repression (minimal medium with 0.4% glucose), or maltose operon induction (minimal medium with 0.4% maltose) (Fig 2A, left panel). RAM2808 cells were grown under conditions of maltose operon repression (minimal medium with 0.4% glycerol as a carbon source), LamB induction (minimal medium with 0.4% glycerol + 0.2% arabinose) or LamB and maltose operon induction (minimal medium with 0.4% maltose + 0.2% arabinose) (Fig 2A, right panel). Having validated these strains, they were used for accumulation study.

### Accumulation and transport specificity
Using these isogenic strains under different expression conditions, the accumulation of maltodextrin conjugates Cpd-1 and

**Cpd-1** (M: 907.93 g/mol)

**Cpd-2** (M: 1394.36 g/mol)

Figure 1.  Chemical structure of the maltodextrin compounds studied in this work.
The moieties corresponding to maltotriose of Cpd-1 and maltohexaose of Cpd-2 are boxed.

Cpd-2 was demonstrated in both cell types and shown to be associated with LamB expression induced by arabinose in RAM2808 or by maltose in RAM1292 as shown in Fig 2B and C. A very weak fluorescence signal was measured in the strain RAM2807, regardless of the sugar added to the culture medium (maltose and /or arabinose), likely corresponding to the nonspecific adsorption of the compounds on the surface of the cells (data not shown). To quantify conjugate accumulation, calibration curves were generated to measure the number of molecules accumulated per bacterial cell (Fig S4).

To follow the intracellular accumulation of conjugates in individual bacterial cells, time-lapse experiments on a deep ultraviolet (DUV) microscope were carried out (Fig 2D and E) using methods previously described (29, 31). The external concentration of Cpd-1 and Cpd-2 was 10 $\mu g \cdot ml^{-1}$ during incubation. Bacterial cells were plated and observed for 30 min. Fluorescence was higher in RAM2808 cells grown in minimal medium incubated with maltosaccharide conjugates only when the cells were induced with arabinose (Fig 2D). Analyses of microspectrofluorimetric data obtained with RAM2808 cells incubated with the two fluorescent

compounds (at 22 $\mu M$) showed the same trend (Fig 2E). Thus, microspectrofluorimetry showed accumulation in RAM2808 cells only when the LamB production was induced. The box-and-whisker plot representation shows the heterogeneity of accumulation in the cells. Similar to the accumulation measured at the population level (Fig 2B and C), individual cells showed more effective accumulation with Cpd-1 relative to Cpd-2 (Fig 2E).

Maltose induction was also able to increase Cpd-1 and Cpd-2 accumulation in the parental strain, RAM1292 (Fig 2B and C), consistent with induction of LamB and MalE expression (Figs 2A and S3). Moreover, the rate of accumulation was studied by using increasing concentrations of Cpd-1 with the parental strain (RAM1292) grown in minimal media with maltose. A plateau was observed with concentrations of Cpd-1 at ~44 $\mu M$ (Fig S5). The presented plots suggest that the uptake of Cpd-1 in *E. coli* cells is saturable at high concentration for 30 min incubation.

### Maltose competition

To evaluate the selectivity of Cpd-1 transport by LamB, we investigated the intracellular accumulation of Cpd-1 in individual

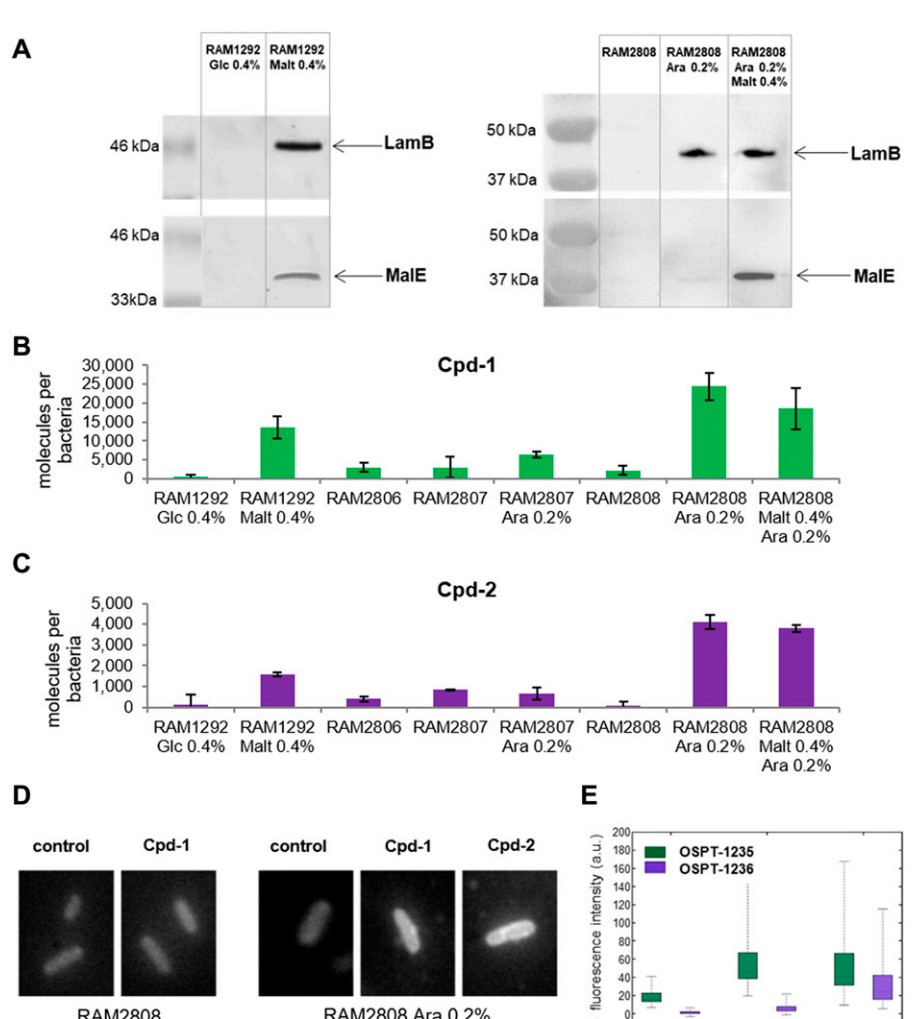

**Figure 2. Accumulation of Cpd-1 and Cpd-2 depends on LamB expression.**
The strains were grown in different media to control the expression of the LamB porin and the MalE transporter; 2 components of the maltose regulon (see Figs S2 and S3). See Fig S1 for the characteristics and the corresponding immunoblots of the strains. **(A)** Presence of LamB and MalE by Western blot in RAM1292 and RAM2808 under different growth conditions. **(B, C)** Number of Cpd-1 (B) and Cpd-2 (C) molecules accumulated per cell in the various studied strains following analysis by spectrofluorimetry. The columns with bars (SDs) correspond to measurements carried out in triplicate. Calibration curves were used to obtain the number of molecules per cell (Fig S4). **(D)** Microfluorimetric images obtained with DUV microscopy with pellets of RAM2808 with or without induction of the LamB porin incubated without and with Cpd-1 or Cpd-2. Controls are RAM2808 cells incubated without Cpd-1 and Cpd-2. **(E)** Microfluorimetric results obtained from (D). Data are represented with a box-and-whisker plots, which is a way of summarizing the essential profile of a quantitative statistical series: the boxes represent data-points from the 25th to 75th percentiles; the middle horizontal lines represent the median data point and the whiskers show the span of the data for each sample. The outliers are represented by red + signs.

bacterial cells in the absence or in the presence of maltose, a natural substrate of the LamB channel (Fig S6). An excess of maltose reduced the uptake of Cpd-1 in *E. coli* RAM1292 cells previously grown under conditions of maltose operon induction, consistent with a maltose regulatory feedback, further supporting the hypothesis that Cpd-1 crosses the outer membrane via LamB channels.

### Time-course accumulation of fluorescently labeled maltodextrin conjugates

#### Bacterial population

To study the time-course of accumulation, RAM2808 cells were treated with either compound for 5–90 min under conditions of maltose operon repression, LamB induction (with arabinose), or LamB and maltose operon induction (0.4% maltose + 0.2% arabinose). Consistent with the results above, relatively low accumulation was detected for

either compound when the maltose operon was repressed (Fig 3A–D). The intracellular content of both Cpd-1 and Cpd-2 under LamB induction and maltose operon induction conditions increased over time and for both compounds were higher than the corresponding uninduced conditions, consistent with the results above (Figs 2 and 3).

#### Individual cells

In a parallel experiment, RAM1292 cells, either under conditions of maltose operon repression (0.4% glucose) or induction (0.4% maltose), were treated with Cpd-1 or Cpd-2 and uptake was measured in individual cells. Fluorescence intensities, corrected for background, photobleaching, and crosstalk (29) corresponding to about 30–40 individual bacterial cell measurements are plotted in Fig 3E. After a short lag of 3–4 min, a roughly linear penetration rate in the maltose-induced RAM1292 cells was observed until roughly 10 min of incubation with the two compounds. Interestingly, Cpd-1 demonstrated

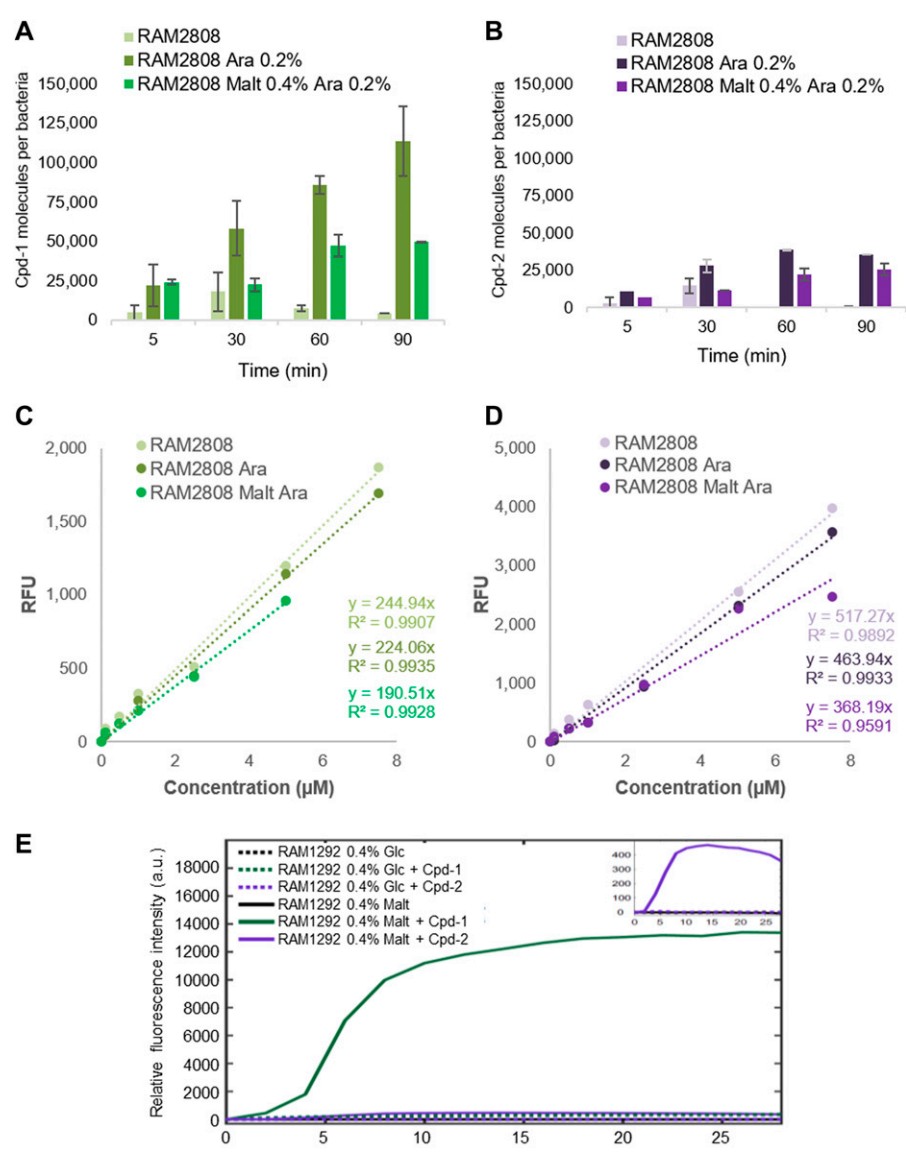

**Figure 3. Time-course of accumulation of Cpd-1 and Cpd-2 in RAM2808 and RAM1292 with repression or induction of the maltose operon.**
**(A, B)** Time-course accumulation in RAM2808 cells grown without or with 0.2% arabinose (Ara, induction of the LamB expression), 0.4% maltose, and 0.2% arabinose (Malt Ara, induction of the maltose operon and LamB expression); and incubated with Cpd-1 (A) or Cpd-2 (B). The columns with bars (SDs) corresponded to measurements carried out in triplicate. **(C, D)** Calibration curves of the fluorescence of Cpd-1 (C) and Cpd-2 (D) used to obtain the number of molecules per cell shown in (A) and (B). **(E)** Time-course accumulation measured with DUV microspectrofluorimetry in RAM1292 cells grown with 0.4% glucose (Glc, repression of the maltose operon, dotted lines) or 0.4% maltose (malt, induction of the maltose operon, full lines). Cell pellets were resuspended extemporaneously under DUV microscope without (black lines) or with Cpd-1 (green lines) or Cpd-2 (purple lines). Insert: Enlargement of the Cpd-2 results.

a rapid increase phase (4–7 min) followed by a steady-state level of accumulation at about 15–18 min. With Cpd-2, a slow increase was observed during the same period under the induced conditions and the obtained steady state was lower (calculated rate for Cpd-1 was 1250 A.U./min versus 75 for Cpd-2). In contrast, no significant accumulation was obtained in the non-induced strain (Fig 3E).

It should be noted that although we cannot precisely determine the correlation between the signal intensity and the number of molecules inside individual bacterial cells (in these conditions, a calibration curve is not possible because of technical conditions, e.g., photobleaching (29, 31)), we were able to observe a relationship between the incubation time and the accumulation rate in individual bacterial cells.

### Interaction of Cpd-1 and Cpd-2 with a single LamB channel

To support the observed LamB-dependent cellular uptake of Cpd-1 and Cpd-2, we exploited electrophysiology to characterize the interaction of fluorophore-conjugated and parent maltodextrins with a single LamB trimer reconstituted in a planar lipid bilayer. This method analyzes the substrate-induced ion current fluctuations across a membrane channel under an applied transmembrane electric field, providing insights on the interaction and, to some extent, the flux of substrates through the channel.

In Figs 4A–D and S7, we show examples of ion current traces in which ion flow (current) was transiently interrupted (blocked) by the interaction of Cpd-1 or Cpd-2 with LamB on either the extracellular (cis) or periplasmic (trans) side of the membrane, as previously observed for malto-oligosaccharides (34). Statistical analyses of these ion flow blockages provided both the association rate ($k_{on}$), obtained from the interaction frequency, and the duration of the interaction, or residence time ($\tau$) (Fig 4E–H and Table S1).

Inspection of the summary of the kinetic parameters allows a qualitative conclusion on the degree to which the small (maltotriose and Cpd-1, Fig 4E and G) or large (maltohexaose and Cpd-2, Fig 4F and H) substrates permeate through the channel or simply bind and bounce back unproductively. Our data showed that the small substrates (shown above to penetrate better than larger substrates through LamB in whole cells) are characterized by short $\tau^{cis}$ (≤130 ± 60 $\mu$s), whereas the large substrates present long $\tau^{cis}$ (≥930 ± 120 $\mu$s) values, in agreement with enhanced interactions of the long sugars with the inner wall of the channel (34). On the other hand, we found that all sugars except for Cpd-2 presented a low $k_{on}^{cis}$ (≤3.0 ± 0.8 × 10$^6$ M$^{-1}$s$^{-1}$) and follow a similar trend on the enhanced trans side ($k_{on}^{trans}$ ≤ 6.3 ± 1.8 M$^{-1}$s$^{-1}$), a behavior consistent with having a lower energy barrier at the trans side of LamB as observed by Danelon et al. (34). In addition, a small effect from electro-osmosis is also observed (most clearly seen with maltotriose). Electro-osmosis is revealed in cation-selective channels, such as LamB, by an enhanced cation current at negative applied voltage ($k_{on}^{cis,M3}$ = 3.0 ± 0.8 × 10$^6$ M$^{-1}$s$^{-1}$) as compared with positive voltage ($k_{on}^{cis,M3}$ = 1.8 ± 0.6 × 10$^6$ M$^{-1}$s$^{-1}$) (35).

In contrast with the other compounds tested here, Cpd-2 has a larger $k_{on}$ (~10 ± 3 × 10$^6$ M$^{-1}$s$^{-1}$). Surprisingly, a shorter overall $\tau$ average is observed for Cpd-2 ($\tau^{trans}$ ~290 ± 60 $\mu$s) as compared with its parent sugar maltohexaose ($\tau^{trans}$ = 725 ± 270 $\mu$s). Our interpretation of this is that Cpd-2 has a substantial contribution of short-lived, unproductive interactions that contribute to decrease $\tau$

as we feel the alternate explanation that Cpd-2 permeates twofold faster than parent maltohexaose is unlikely. This effect is more clearly seen on the trans side, consistent with an interpretation that the lower energy barrier at the periplasmic opening offers the substrate more opportunities of binding and exit through the same side. The fluorophore may constrain the helical conformation needed to screw the molecule across the channel, preventing it from exiting through the extracellular side. This may not be the case for Cpd-1, where the requirement of a pronounced structural rearrangement of the sugar may be neglected given the small size of the sugar, allowing the molecule to diffuse through the channel with $\tau$ values similar to those of maltotriose (M3 in equation)

$$\left(\tau^{cis,M3} \approx 90 \pm 30\,\mu s;\ \tau^{cis,Cpd-1} \approx 110 \pm 40\,\mu s\right).$$

Accordingly, our interpretation of this data is that Cpd-1 can permeate LamB with a similar rate as the maltotriose sugar alone, whereas diffusion of Cpd-2 is sterically hampered relative to maltohexaose.

Using a rough approximation of the molecular flux ($\phi = k_{on}$ [c]) suggests that at 10 $\mu$M concentration ~1,000 Cpd-1 molecules permeate per second and per LamB monomer. This is in overall agreement with the trends of the whole-cell penetration data shown above wherein Cpd-1 accumulates more readily relative to Cpd-2. Moreover, analysis of the electrophysiology data suggests a molecular specificity for Cpd-1 versus Cpd-2 with respect to interaction and permeation. This potentially suggests a limitation due to size or diffusion efficiency inside the LamB channel for this approach (34, 36).

### Time-course and subcellular localization of Cpd-1 and Cpd-2

Having established that both Cpd-1 and Cpd-2 can pass the outer membrane via LamB, questions remained concerning their cellular localization and disposition inside the bacterial cell. Maltodextrins are generally transported into the bacterial cytoplasm across the outer and inner membranes, via LamB/MalE and maltose inner membrane transporters (MalF,G,K), eventually to be metabolized by cytoplasmic enzymes (MalP,Q,Z) (Fig S2) (23). RAM1292 cells, primed for either repression or induction of the maltose operon, were incubated with Cpd-1 as above and the bacterial cells were collected at various times. Then, an adapted fractionation procedure (37) was applied to separate the periplasmic fluid, the cytoplasm, and a crude membrane fraction comprising outer and inner membranes to allow measurement of Cpd-1 and its localization (Figs 5 and S8). The distribution of LamB-, MalE-, and EfTu-specific proteins of the three different cellular fractions (membrane, periplasm, and cytoplasm, respectively) was confirmed by immunodetection (Fig S8). The total amount of Cpd-1 detected in the three cellular fractions increased during the incubation times, with the most of the Cpd-1 signal being found in the periplasm at all time-points (Fig 5).

### Behavior of maltosaccharide conjugates inside the cell

To understand the fate of Cpd-1 inside the cell, RAM2808 cells were incubated in the presence of Cpd-1 corresponding to a

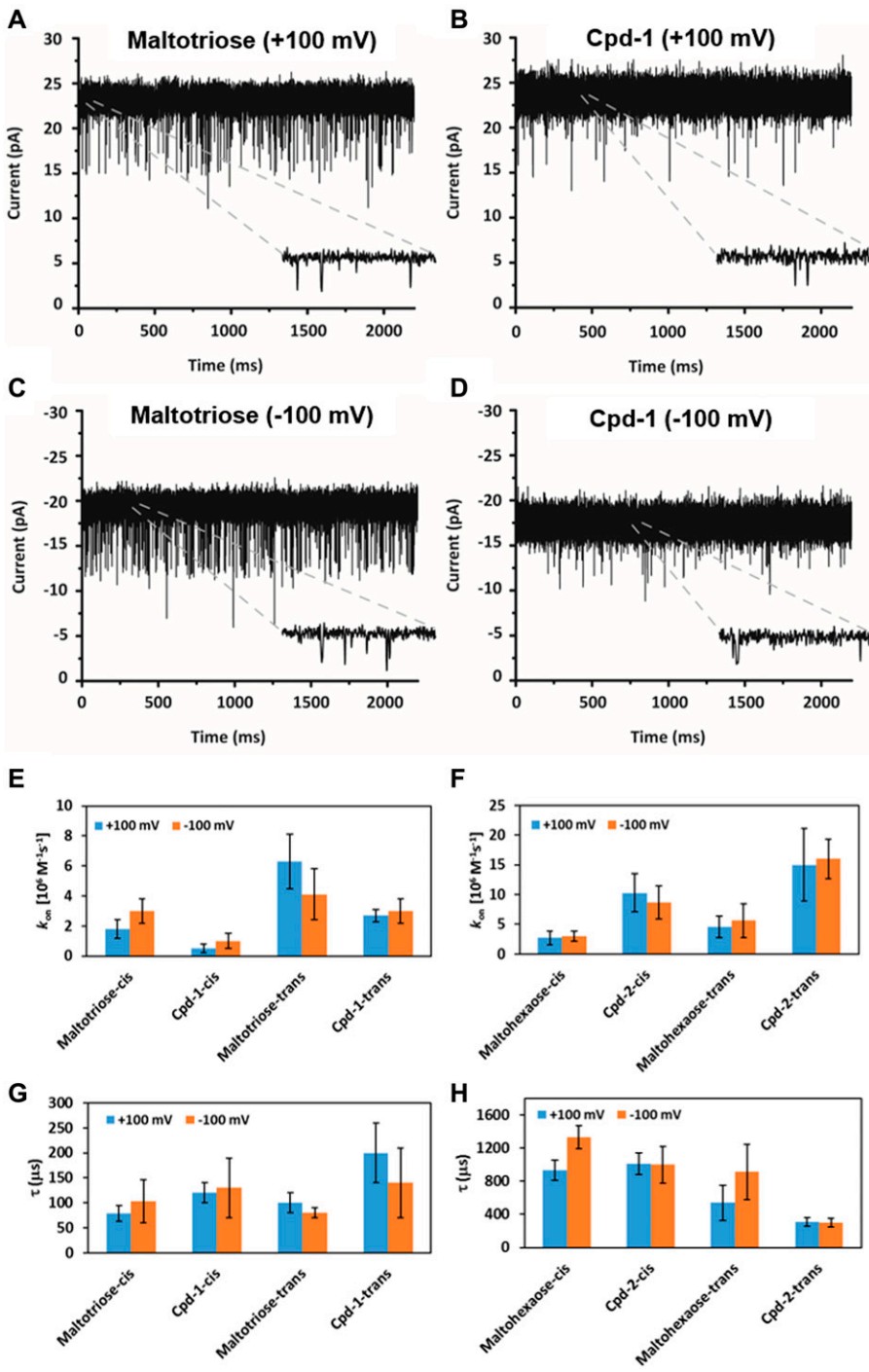

**Figure 4. Interaction of maltotriose, maltohexaose, Cpd-1, and Cpd-2 with LamB.**
**(A–D)** Cpd-1 interacts similarly to maltotriose with LamB. **(A, B)** Ion current recordings showing interaction of LamB on cis-side addition of 10 μM maltotriose (A) and 10 μM Cpd-1 (B) in 1 M KCl 10 mM Hepes, pH 7, on application of + 100 mV. **(C, D)** Ion current recordings showing interaction of LamB on cis-side addition of 10 μM maltotriose (C) and 10 μM Cpd-1 (D) in 1 M KCl 10 mM Hepes, pH 7, on application of −100 mV. For Cpd-2, see Fig S7. Insets show the zoomed-in view of the traces showing single interaction event. **(E–H)** Analysis of the ion current fluctuations reveals the kinetic parameters of the sugar interaction with LamB. **(E, F)** The on-rate ($k_{on}$) for maltotriose or Cpd-1 (E) and for maltohexaose or Cpd-2 (F) addition at cis and trans side addition. **(G, H)** The residence time ($\tau$) for maltotriose or Cpd-1 (G) and for maltohexaose or Cpd-2 (H) inside the channel after addition at cis- and trans-side addition. Note: In each experiment, we applied ± 100 Mv. As LamB is cation-selective positive voltage, it causes an ion flow from cis to trans, whereas negative voltages cause an opposite flow.

"pulse time," and they were then pelleted by centrifugation and resuspended in fresh medium without Cpd-1. Samples were removed at various time points (a "chase period") in analogy to pulse-chase assays performed with radiolabeled compounds, and the intracellular level of Cpd-1 was determined (Figs 6 and S9). A decrease of the Cpd-1 fluorescence signal during the chase period was observed. The level of decrease was similar in the absence or presence of carbonyl cyanide m-chlorophenylhydrazone (CCCP)

which collapses the energy-driving force of the efflux pump (8, 15) (Fig 6A). This suggests that efflux pump activity does not significantly alter the intracellular accumulation of Cpd-1 and efflux is not a driving factor in the observed decrease of signal in these experiments (Fig 6A).

Importantly, when the bacterial cells are grown in the presence of maltose to fully induce the maltose regulon, a reduction of the intracellular Cpd-1 was obtained relative to cells induced only with

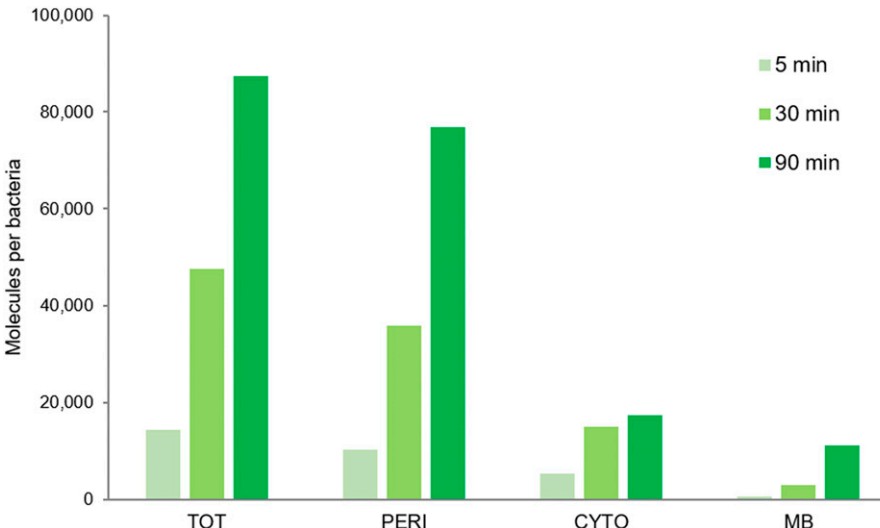

**Figure 5.  Detection of Cpd-1 in the different cell compartments of RAM1292.**
The RAM1292 cells were grown in the presence of 0.4% maltose and incubated with Cpd-1. Samples were recovered at various time points and a fractionation protocol was performed to obtain the total (TOT, periplasm + spheroplast), periplasmic (PERI), cytoplasmic (CYTO), and membrane (MB) fractions of the cells. Fluorospectrometry measurements were performed to determine the levels of Cpd-1 in each fraction. Note: Western blots were performed to confirm the distribution of specific proteins of the different cellular fractions (Fig S8A). Calibration curves were used to obtain the number of molecules per cell and per compartment (Fig S8B).

arabinose (Fig 6B). This phenomenon can also be seen above in Figs 2B and 3A. Indeed, the intracellular concentration of Cpd-1 in RAM2808 cells induced with only arabinose is higher than that in the RAM2808 induced with arabinose and maltose. One possible explanation for this could be that, in the maltose-induced bacteria, the complete metabolic degradative pathway is fully present and active and is therefore able to cleave/metabolize the maltose portion of the molecule, although it is unclear how this would necessarily lead to the reduction of signal related to the perylene component (Figs 6 and S2).

### Cpd-1, like maltose, can induce the maltose operon

Having demonstrated that Cpd-1 accumulates in all three cellular fractions (Fig 5), we wanted to investigate whether the accumulation of Cpd-1 in the cytoplasm is able to induce the expression of the maltose regulon similar to maltose and maltodextrins. For example, is the cytoplasmic concentration of Cpd-1 sufficient to induce the expression of *malE* and to ensure the production of maltose-binding protein (MalE) in the periplasmic space (Fig S2). To study this, we measured the levels of MalE in the periplasmic fraction of RAM2808 cells over time following incubation with Cpd-1 and inducing sugars (Fig 7). When RAM2808 cells were grown under arabinose conditions (induces LamB only), incubation with Cpd-1 was shown to induce MalE over 5–60 min (Fig 7), consistent with the fractionation data showing cytoplasmic accumulation of Cpd-1 (Fig 5) and activation of the mal operon located in the cytoplasm (Fig S2).

## Discussion

A key challenge in the discovery and optimization of antibiotics against Gram-negative bacteria is translocation of the antibiotics across the two bacterial membranes leading to sufficient accumulation inside the cell (kinetics, location, and concentration) to exert the desired effect on its molecular target (4, 5, 11, 12). Several

approaches have been explored to bypass these barriers including new adjuvants (permeabilizer, efflux pump inhibitors) to rejuvenate the antibacterial efficacy, Trojan horse strategies to hijack active transport pathways and provide efficient penetration, etc. However, crucial points with these strategies concern the internal biological activity and the precise localization of the molecule, and these aspects are generally not studied or reported in depth (25, 26).

Our aim in the present study was to quantify the intracellular accumulation and subcellular localization of maltodextrin–fluorophore conjugates and determine the potential of such maltodextrins to assist in membrane translocation. By using protocols recently developed (11, 31), the cellular distribution in bacterial compartments and the behavior of the compounds were analyzed by using spectrofluorimetry in bacterial populations and microfluorimetry in individual bacterial cells. The travel of maltodextrin derivatives across the LamB channel reconstituted in artificial membranes was also investigated.

By following the accumulation in bacterial populations and in individual bacterial cells, we demonstrated that the uptake of Cpd-1 and Cpd-2 is related to the expression of the LamB porin in the outer membrane. In addition, we have shown that Cpd-1 shows higher accumulation compared with Cpd-2 under similar conditions. In parallel, data obtained from electrophysiology studies were consistent with these conclusions wherein Cpd-1 penetrates through reconstituted LamB channels better than Cpd-2. These general results wherein shorter saccharides provide improved accumulation are in agreement with recent results suggesting that a thiomaltose–perylene conjugate has better accumulation relative to a maltohexaose–perylene conjugate (26).

Particularly, we determined the distribution of Cpd-1 inside the bacterial cell to follow the steps involved during the uptake pathway. First, we demonstrated that Cpd-1 is able to use the LamB channel to pass across the OM barrier by using electrophysiology and microspectrofluorimetry methods. Second, Cpd-1 was detected both in the periplasm and in the cytoplasm, with minor amounts associated with mixed-membrane fractions and all fractions

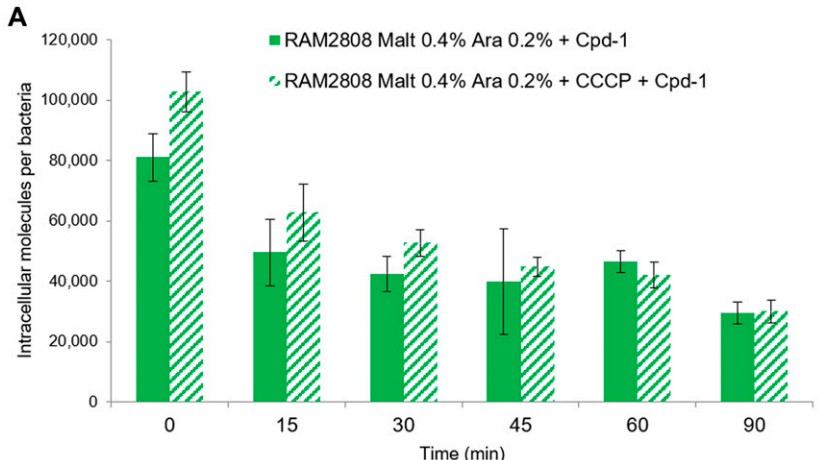

**Figure 6. Behavior of Cpd-1 signal in an incubation chase study.**
RAM2808 cells were grown in the absence or presence of 0.4% maltose and induced with 0.2% arabinose (Ara, induction of LamB expression; and Malt Ara, induction of the maltose operon and LamB expression). The bacterial suspension was then incubated with Cpd-1 in the absence or presence of carbonyl cyanide m-chlorophenylhydrazone (CCCP, which collapses the energy driving force of efflux pumps). **(A)** After incubation of the RAM2808 Malt Ara cells, samples were centrifuged, and cell pellets were resuspended in the same buffer without any Cpd-1. At 0, 15, 30, 45, 60, and 90 min, samples were recovered and analyzed. The columns with bars (standard deviations) corresponded to measurements carried out in triplicate. **(B)** To compare the degradation of Cpd-1 in cells according to the regulation of the maltose operon (induced or not, Malt Ara or Ara conditions, respectively), the % of signal was standardized from 0 time and the degradation level (in percent) was reported at 30 and 60 min in RAM2808 cells grown with (induction of the maltose operon) or without maltose (repression of the maltose operon) (means of three independent assays performed in triplicates). Calibration curves were used to obtain the number of molecules per cell (Fig S9).

| | | RAM2808 + Ara 0.2% | RAM2808 + Ara 0.2% + Malt 0.4% |
|---|---|---|---|
| **Expression** | | LamB | All maltose regulon members (see Fig S2) including: (LamB, MalE, MalF, MalG, MalK) Maltose degradation enzymes (MalS, MalP, MalQ, MalZ) |
| **Incubation time** | 30 min | 33% (±4) | 42% (±4) |
| | 60 min | 40% (±12) | 53% (±8) |

showing an increase in signal over longer incubation times. It is worthwhile mentioning that the amount of Cpd-1 in the cytoplasm is low compared with that observed in the periplasm. This could be because of poor transit across the inner membrane, internal degradation, or alternatively an expel *via* efflux pumps such as AcrAB-TolC. During the incubation of pre-charged bacterial cells, a noticeable decrease in the total signal was observed (about 50% over 30 min) and the addition of CCCP did not significantly increase the amount detected. This suggests that the RND-efflux pumps such as AcrAB, which are inhibited by the protonophore CCCP, are not involved in the decrease of internal amount of Cpd-1. Interestingly, when the bacteria are cultivated in the presence of maltose to fully induce the maltose regulon, a reduction of accumulation is observed which may support a role of inner membrane transporters and/or maltodextrin metabolism machinery in the reduction of intracellular Cpd-1 observed in the "pulse-chase" experiment. This clearly demonstrated that this maltotriose conjugate is able to reach the cytoplasm where it could be degraded by maltohydrolases and etherases present in this compartment (23, 38).

Importantly, it is worthwhile to note that the cytoplasmic concentration of Cpd-1 is able to trigger activation of maltose regulon and the expression of MalE, a component involved in maltodextrin transport. This is a key characteristic, because the maltodextrin conjugate is able to activate the assembly of its proper "translocation pathway" to the cytoplasm. We propose that Cpd-1 can promote its self-uptake by up-regulating the cellular machinery involved in the translocation pathway.

To conclude, these studies demonstrated that the maltodextrin conjugate Cpd-1 penetrates the outer membrane barrier of *E. coli* and accumulates in the cytoplasm where it is able to activate the genetic cascade of the maltose regulon. This more complete characterization of the uptake of such maltosaccharide conjugates opens the way to potential exploitation of the maltose uptake pathway for delivering payloads straight inside bacterial cytoplasm.

## Materials and Methods

### Strains and media

The bacteria used in this study are listed in Fig S1. Bacteria RAM1292 (WT strain) and its derivatives RAM2806 (LamB-), RAM2807 (LamB- with empty pBAD24 plasmid) and RAM2808 (LamB- with pBAD24 plasmid encoding *lamB* under ara promoter) were a generous gift from Dr Misra (33, 39). Bacteria were grown in Luria–Bertani broth (LB: 10 g tryptone, 5 g yeast extract, 10 g NaCl, per L; pH 7) and when required, ampicillin (Amp, 25 $\mu$g·ml$^{-1}$) and/or chloramphenicol (Cm, 20 $\mu$g·ml$^{-1}$) were added.

RAM1292 and RAM2808 bacteria were serially passaged over a period of 3 d, to enrich and to pre-condition the cells to the desired growth conditions. First day: A colony was inoculated in LB (+Amp + Cm) and incubated 24 h at 37°C with shaking at 160 rpm. Second day: LB pre-cultures were inoculated at 1:250 in minimal medium M9

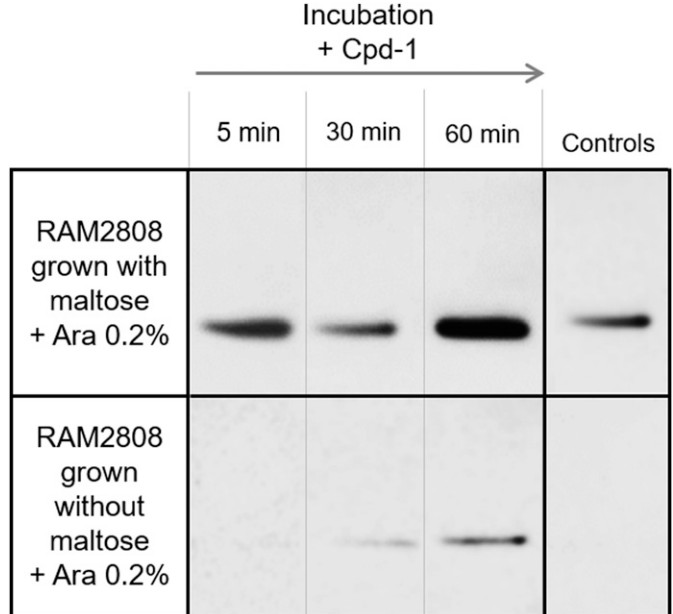

**Incubation + Cpd-1**

| | 5 min | 30 min | 60 min | Controls |
|---|---|---|---|---|
| RAM2808 grown with maltose + Ara 0.2% | | | | |
| RAM2808 grown without maltose + Ara 0.2% | | | | |

**Figure 7. Induction of MalE expression by Cpd-1 in RAM2808.**
RAM2808 cells were grown with (top) or without (bottom) 0.4% maltose and induced with 0.2% arabinose (Ara, expression of LamB). The bacterial suspensions were incubated with Cpd-1. Samples were recovered at various time points and a cellular fractionation protocol was performed. Controls are the same bacterial suspensions incubated without Cpd-1 recovered at 30 min. Presence of MalE in the periplasmic fraction was monitored over time by Western blot analysis.

(6.78 g $Na_2HPO_4$, 3 g $KH_2PO_4$, 1 g $NH_4Cl$, and 0.5 g NaCl per L) complemented with 2 mM $MgSO_4$, 100 $\mu$M $CaCl_2$, 0.1 ‰ vitamin B1 (vol/vol) and 0.2% casamino acid (vol/vol) in which 0.4% (wt/vol) glucose was added (+Amp + Cm) and incubated 24 h at 37°C with shaking at 160 rpm. Third day: M9 pre-cultures were centrifuged at 3,000 $g$ for 15 min at 20°C and the cell pellets were resuspended in M9 to obtain an $OD_{600}$ around 1.6. Fresh M9 broth was inoculated at 1:50 with the $OD_{600}$ 1.6 solution and incubated 24 h at 37°C with shaking at 160 rpm. To induce the maltose operon or not, 0.4% maltose or glucose were added, respectively, in the RAM1292 cultures, and 0.4% maltose or glycerol in the RAM2808 cultures (+Amp + Cm) (the LamB- RAM2808 cells do not need to be repressed with glucose, but glycerol is added as a carbon source to allow better growth of the bacteria in a minimal medium). On the day of the experiment, pre-cultures were diluted in fresh broth cultures to obtain an $OD_{600nm}$ of around 0.1. After 2 h of growth ($OD_{600}$ ~ 0.4), cells carrying the pBAD24-$lamB$ plasmid were induced with 0.2% arabinose for another 2 h.

### Accumulation assay

Bacterial cells were grown in M9 minimal medium (see the Strains and media section) and supplemented with different carbon sources as appropriate. In our assays, glucose was used to repress the expression of maltose operon (in strain RAM1292), arabinose was used to induce the expression of LamB (in strain RAM2808), and maltose was used to induce the expression of maltose operon (in strains 1292 and 2808).

After growth, bacteria were centrifuged at 3,000 $g$ for 25 min at 20°C. Cells were washed with complemented M9 and centrifuged at 3,000 $g$ for 25 min at 20°C. Pellets were then resuspended in 1:10 of the initial volume in complemented M9 in which 0.01% Tween 20 was added (M9T buffer) to obtain a density of $6 \times 10^9$ $CFU·ml^{-1}$. Bacterial suspensions were incubated in the dark for 30 min in the presence of 100 mM ATP at 37°C in the presence of Cpd-1 and Cpd-2. Bacterial suspensions incubated without Cpd-1 and Cpd-2 were used as controls.

### Spectrofluorimetry and microspectrofluorimetry

In the accumulation assay, samples were removed at various time points and cell suspensions (800 $\mu$l for spectrofluorimetry measurement or 400 $\mu$l for microspectrofluorimetry) added to M9T (1,100 $\mu$l or 550 $\mu$l, respectively) and centrifuged at 9,000 $g$ for 10 min at 4°C to eliminate extracellular adsorbed compounds. Bacterial cell pellets were washed with M9T and a second wash was performed with M9 medium to eliminate the Tween 20.

#### *Spectrofluorimetry measurements*
After centrifugation, pellets of the washed cell suspensions (corresponding to 800 $\mu$l of bacterial suspensions) were resuspended in 500 $\mu$l of 0.1 M glycine-HCl (pH 3) and lysed overnight at room temperature. Following 15 min centrifugation at 15,000 $g$ at 4°C, lysates were deposited into 96-well black microplate and fluorescence measured at Ex 260 nm (Infinite Pro microplate reader; Tecan). The emission peak was around 450 nm. The fluorescence signals were corrected by the tryptophan peak of the bacteria to obtain a fluorescence signal per bacterial cell (31). Calibration curves were carried out to determine the quantity of molecules accumulated per cell (31): various concentrations of Cpd-1 and Cpd-2 were mixed with bacterial lysates at $OD_{600}$ of 4.8 and measured using spectrofluorimeter (n = 3 replicates).

#### *Microspectrofluorimetry measurements*
To detect the Cpd-1 fluorescence from single bacteria background, pellets corresponding to 400 $\mu$l of RAM2808 bacterial suspensions were resuspended in 40 $\mu$l of M9T. 0.5 $\mu$l of resuspended pellets were deposited between two quartz coverslips and analyzed by DUV fluorescence imaging (Ex 260 nm and Em 420-480 nm) at DISCO Beamline (40).

### SDS–PAGE and Western blot analysis

Protein samples (obtained in the fractionation section) solubilized in loading buffer (41) were heated 15 min at 95°C to totally denature the various protein complexes and samples. The same amounts of protein samples (checked by Coomassie blue staining 0.2 $OD_{600}$ units) were separated on SDS-polyacrylamide mini gels (10% polyacrylamide, 0.1% SDS). Proteins were electro-transferred onto nitrocellulose membranes in transfer buffer (0.05% SDS, 20 mM Tris, 150 mM glycine, 20% isopropanol). An initial saturating step was performed overnight at 4°C with Tris-buffered sodium (TBS: 50 mM Tris–HCl, pH 8.0, 150 mM NaCl) containing skimmed milk powder (4%). The nitrocellulose sheets were then incubated in TBS +

skimmed milk powder + Triton X-100 (0.2%) for 1 h at room temperature in the presence of polyclonal antibodies directed against LamB, MalE, EfTu, OmpC, and OmpF proteins (1:10,000, 1:5,000, 1:30,000, 1:10,000, and 1:10,000, respectively). After washes, the detection of antigen–antibody complexes was performed with horseradish peroxidase–conjugated Immune-Stargoat anti-rabbit IgG antibodies (Bio-Rad) and detection was performed with the Clarity Western ECL kit (Bio-Rad) using a Chemidoc XRS+ (Bio-Rad).

## Time-course accumulation in individual bacterial cells

The time-course accumulation in individual bacterial cells was studied by using a previously described protocol (29, 30). Bacterial cells were grown in complemented M9 supplemented with different carbon sources as appropriate. Bacteria were concentrated to obtain an $OD_{600}$ of 4.8 in M9T. Then, 120 $\mu l$ of the suspension were centrifuged at 6,000 $g$ for 10 min at 4°C. The pellets were resuspended extemporaneously in 40 $\mu l$ of M9T containing or not containing Cpd-1 or Cpd-2. 0.5 $\mu l$ of resuspended pellets were deposited between two quartz coverslips and analyzed by DUV fluorescence imaging at DISCO Beamline. The accumulation in individual bacterial cells was monitored directly under a DUV microscope during 30 min. Bacterial cells were first located in brightfield before excitation in DUV through a microscope (Zeiss Axio Observer Z-1) at Synchrotron SOLEIL (42). Emission was collected via a Zeiss ultrafluar objective at 100× with glycerin immersion. The fluorescence was recorded by exciting at 260 nm through an emission bandpass filter at 420–480 nm (OMEGA Optical, Inc). For the tryptophan fluorescence, the emission was collected through an emission bandpass filter at 327–353 nm (SEMROCK). Fluorescence images were recorded with a BUV CMOS (Prime 95B; Photometrics). The whole setup (microscope, stages, shutters, filters, and camera) was controlled by Micro-Manager (43). Bacteria were observed for 30 min with a sampling time of 2 min for each area: 30 s with the filter 1 (Cpd-1 and Cpd-2 molecules fluorescence detection), 30 s with the filter 2 (tryptophan fluorescence detection), followed by 60 s of pause. During the intervening pause in sampling, the same acquisition cycle was performed on another field of view, avoiding constant UV irradiation of the same field.

The images were analyzed with Image J (Rasband, W.S., Image J, U.S. National Institutes of Health, Bethesda, MD, USA, http://imagej.nih.gov/ij/) (44). The illumination heterogeneities were corrected before background subtraction. First, the threshold was automatically adjusted using a triangle algorithm; thereafter, bacteria were analyzed as the remaining particles. The mean intensity coming from each bacterium was automatically calculated considering its pixel area. Finally, all bacteria signal taken from one image were averaged. For each condition, seven different localizations with minimum 30 bacteria per field of view were recorded and averaged.

## Electrophysiology

Single-channel electrophysiology measurements were performed using the Montal and Mueller technique (45). In short, a 25-$\mu m$-thick Teflon film containing an aperture of approximately 100 $\mu m$ diameter is sandwiched between the chambers of Teflon cuvette. A solution of 1% (vol/vol) hexadecane in hexane was used to make the area around the aperture more hydrophobic to form stable

bilayers. Membranes were formed using a solution of 5 $mg \cdot ml^{-1}$ DPhPC in pentane. The chambers were filled with 1 M KCl, 10 mM Hepes, pH 7, that serves as an electrolyte. A pair of Ag/AgCl electrodes (World Precision instruments) was used to measure electric current. One electrode is connected to the ground (*cis* side) and the other electrode is connected to the headstage (*trans* side) of the Axopatch 200B amplifier (Axon instruments). Protein was purified using standard extraction protocol (35). Purified protein solubilized in 1% Genapol is added to the *cis* side of the membrane where it reconstitutes spontaneously with the extracellular loops oriented towards the side of protein addition, as verified by a slight voltage-dependent asymmetry of the ion current (Figs 4 and S7) previously observed in other experiments (34, 46). Current measurements were made using Axopatch 200B amplifier in the voltage clamp mode and was digitized using Axon Digidata 1440A digitizer controlled by pClamp software. The current traces were filtered using an analogue low-pass Bessel filter of 5 kHz and sampled at 20 kHz and recorded onto a computer hard drive. Clampfit 10.6 program was used for data analysis. The traces were filtered at 2 kHz and plotted using origin. The association rate and dissociation rate were calculated using single-channel analysis (34, 35).

To analyze the ion current fluctuation with respect to interaction, we use a two-barrier one–binding site model (34):

$$\text{open channel + maltose} \underset{k_{off}}{\overset{k_{on}}{\rightleftarrows} } \text{closed channel} \underset{k_{off}}{\overset{k_{on}}{\rightleftarrows}} \text{open channel}$$
$$\text{+ maltose.}$$

For further analysis, we should note that in our experiment we work under dilute concentrations and thus the number of entries is in good approximation, proportional to the concentration of the maltose in solution. The on-rate $k_{on}$ (association rate) is obtained by counting the number of ion current blockage events per time $n$ [$s^{-1}$] divided by the concentration [c] of the sugar (34). Furthermore, LamB is trimeric and the ion current blockages may originate independently from one of the three monomers:

$$k_{on} = n/3[c]$$

On the other hand, the re-opening is a statistical event and correlated to the strength of the binding. A channel which is closed at t = 0 will have the probability R(t) to open at time t. Within the previously described simple binding model, this leads to an exponentially decaying function R(t) = $e^{-t/\tau}$. Fitting the life-time distribution of a closed channel by an exponential parameter $\tau$ (called residence time) yields the off-rate (dissociation rate) $k_{off} = 1/\tau$ (47).

## Maltose competition in individual bacterial cells

The same protocol used in the time-course accumulation assay was followed with some modifications. Bacterial cells, RAM1292, were grown in M9 (see the Strains and media section) complemented with 0.4% maltose. Bacteria were concentrated to obtain an $OD_{600}$ of 4.8 in M9T. Then, 120 $\mu l$ of the suspension was centrifuged at 6,000 $g$ for 10 min at 4°C. The RAM1292 pellets were resuspended extemporaneously in 40 $\mu l$ of M9T containing or not

containing Cpd-1 (22 $\mu$M) with or without maltose 10× (220 $\mu$M). Fluorescence images were recorded with a BUV EM CCD microscope (Princeton PIXIS 1024 BUV) controlled by Micro-Manager (43). Bacteria were observed for 30 min with a sampling time of 2 min for each area: 7 s with the filter 1 (Cpd-1 molecule fluorescence detection, 435-455), 7 s with the filter 2 (tryptophan fluorescence detection, 327–353). The same acquisition cycle was performed on other fields of view, avoiding constant UV irradiation of the same area. For each condition, four different localizations with minimum 30 bacteria per field of view were recorded and averaged.

### Fractionation protocol

The cells were grown in complemented M9 (see the Strains and media section). RAM1292 was grown with maltose (induction maltose operon) (Fig 5), and RAM2808 was grown without or with maltose 0.4% (induction maltose operon) and with 0.2% arabinose for inducing LamB expression (Fig 7). Bacteria were centrifuged at 3,000 $g$ for 25 min at 20°C and washed with M9T before centrifugation (3,000 $g$, 25 min, 20°C). Pellets were resuspended in 1:10 of the initial volume in M9T to obtain a density of $6 \times 10^9$ CFU·ml$^{-1}$. In flasks and in the dark, the bacterial suspension was incubated for 5, 30, and 90 min at 37°C with Cpd-1 at 55 $\mu$M (Fig 5) or for 5, 30, and 60 min at 37°C with Cpd-1 at 22 $\mu$M (Fig 7) in presence of 100 $\mu$M ATP. Bacterial suspensions incubated without Cpd-1 and Cpd-2 were used as controls.

For the fractionation, a lot of bacteria were needed to observe the fluorescence signal in the different compartments. For that, five pellets were collected and pooled together at the fractionation step. Suspensions (5 × 800 $\mu$l) were loaded on M9T (5 × 1,100 $\mu$l) and centrifuged at 9,000 $g$ for 10 min at 4°C to eliminate extracellular adsorbed compounds and collected washed bacteria. A first wash was performed with M9T and a second wash was performed with M9 medium to eliminate the Tween 20.

After centrifugation, the fractionation protocol was adapted from Pagès et al (37). On ice: the 5 pellets corresponding to 800 $\mu$l of bacterial suspensions were resuspended in a final volume of 750 $\mu$l of 200 mM Hepes HCl, pH 7.4. Then, 750 $\mu$l of 200 mM Hepes, pH 7.4, and 1 M sucrose were added with 75 $\mu$l of 100 mM EDTA, pH 6. Gentle homogenization was performed before adding 1,500 $\mu$l of 500 $\mu$g·ml$^{-1}$ lysozyme in deionized water (ratio: almost $6 \times 10^9$ bacteria/ml). The mixture was incubated 20 min on ice with gentle homogenizations. Then, 150 $\mu$l of 500 mM MgCl$_2$ were added to stabilize the inner membrane. After 5 min of incubation on ice, 500 $\mu$l of the mixture was sampled to represent the total amount of Cpd-1 accumulated in bacteria (Tot). Then, the mixture was centrifuged for 5 min at 9,000 $g$ and 4°C. The supernatant corresponded to the periplasm (Peri), and the pellet to the spheroplasts and the outer membrane.

The pellet was put, 3 times, on dry ice and after in hot water (56°C) to induce a thermal shock before resuspending in deionized water containing 500 $\mu$g·ml$^{-1}$ lysozyme. Spheroplasts were broken by cell disruption (OneShot, CellD) at 2 kBar. The lysate was ultracentrifuged at 150,000 $g$ for 1 h at 4°C. The supernatant corresponded to the cytoplasm (Cyto) and the pellet to the membranes (Mb). The membranes were resuspended in 125 $\mu$l of deionized water.

For the fluorescence detection of Cpd-1 in different cellular compartments (Fig 5), the solution corresponding to periplasm, cytoplasm, and membranes were diluted with the different used buffer to be able to compare the Cpd-1 accumulation in the different compartments and in total cells. The ratio 1:1 with one volume of water for one volume of mix solution (100 mM Hepes, pH 7.4, 500 $\mu$M sucrose, 25 mM MgCl$_2$, and 2.5 mM EDTA) was used. The samples were analyzed by spectrofluorimetry (Ex 260 nm, Em 450 nm, Infinite Pro microplate reader [Tecan]). The spectra were normalized with the tryptophan peak around Em 325 nm before subtraction of control spectra. Standard curves were obtained for the different compartments with the corresponding solution of bacteria without Cpd-1.

To confirm the distribution of specific proteins in the cellular compartments (Fig S8) and to monitor the presence of MalE in the periplasmic compartment (Fig 7), Western blots were performed. For membrane extracts, denaturing buffer (41) was added in 1:1 (vol:vol). For Tot, periplasmic, and cytoplasmic proteins, the volumes of sample were high so a precipitation of proteins with absolute EtOH 1:1 (vol:vol) was performed at –20°C overnight. Then, samples were centrifuged at 15,000 $g$, 20 min at 4°C and the supernatant was removed. Samples were dried in speed vacuum and pellets were resuspended in 50 $\mu$l of denaturing buffer.

### Stability assays

Bacterial cells, RAM2808, were grown in complemented M9 (see the Strains and media section) with 0.4% maltose and induced with 0.2% arabinose for expressing LamB. After centrifugation, cells were washed, and the bacterial suspension concentrated at OD$_{600}$ around six in M9T was incubated 30 min with Cpd-1 (50 $\mu$M) in presence of 100 mM ATP. Then, the suspension was loaded on M9T and centrifuged at 3,000 $g$ for 10 min at 4°C. A first wash was performed with M9T and a second wash was performed with M9 medium to eliminate the Tween 20. The pellet was resuspended in M9 without or with the efflux blocker CCCP used at 10 $\mu$M that collapses the energy-driven force needed by the efflux pump. At 0, 15, 30, 45, 60, and 90 min, samples were collected (800 $\mu$l on 1,100 $\mu$l M9) and centrifuged (9,000 $g$ at 4°C for 10 min) before lysing cells with 500 $\mu$l of 0.1 M Glycine-HCl, pH 3, overnight at room temperature. After a centrifugation for 15 min at 15,000 $g$ at 4°C, 50 $\mu$l of lysates were deposited into 96-well black microplate and the fluorescence was measured at Ex 260 nm (Infinite Pro microplate reader; Tecan). The emission peak was around 450 nm. Calibration curves were carried out to determine the quantity of molecules accumulated per cell. Various concentrations of Cpd-1 were mixed with bacteria lysates at OD$_{600}$ = 4.8 and measured with spectrofluorimeter (n = 3).

## Supplementary Information

## Acknowledgements

We gratefully acknowledge R Misra for his generous gifts of *E. coli* strains. We thank Anne Davin-Regli and Muriel Masi for their fruitful discussions, Anne-

Marie Tran, Valérie Rouam, and Blandine Pineau for technical assistance during the assays and Pan Chan, Steve Baker, and Steve Rittenhouse for critical reading of the manuscript. The research leading to these results was conducted as part of the TRANSLOCATION and ENABLE consortia, and it has received support from the Innovative Medicines Initiatives Joint Undertaking under Grant Agreement no. 115525 (TRANSLOCATION) and no. 115583 (ENABLE) resources which are composed of financial contribution from the European Union's seventh framework program (FP7/2007–2013) and European Federation of Pharmaceutical Industries and Associations companies in kind contribution. This work was also supported by Aix-Marseille University and Service de Santé des Armées, and by Soleil program (projects no. 20151274, 20160173, 20160883, 99170096M).

## Author Contributions

E Dumont: data curation, formal analysis, investigation, methodology, and writing—original draft.
J Vergalli: data curation, formal analysis, investigation, methodology, and writing—original draft.
J Pajovic: data curation, investigation, and methodology.
SP Bhamidimarri: data curation, investigation, and methodology.
K Morante: investigation and methodology.
J Wang: investigation and methodology.
D Lubriks: investigation and methodology.
E Suna: investigation and methodology.
RA Stavenger: conceptualization, supervision, project administration, and writing—original draft, review, and editing.
M Winterhalter: conceptualization, supervision, project administration, and writing—original draft, review, and editing.
M Réfrégiers: conceptualization, software, supervision, and writing—original draft, review, and editing.
J-M Pagès: conceptualization, supervision, project administration, writing—original draft, review, and editing.

## Conflict of Interest Statement

The author (RA Stavenger) declares competing financial interests: RA Stavenger is an employee and shareholder of GlaxoSmithKline.

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
