## [Reviewer comments · Life Science Alliance]

Life Science Alliance

Mechanistic aspects of maltotriose-conjugate translocation to the Gram-negative bacteria cytoplasm

Jean-Marie Pages, Estelle DUMONT, Julia VERGALLI, Jelena PAJOVIC, Satya BHAMIDIMARRI, Koldo MORANTE, Jiajun WANG, Dmitrijs LUBRIKS, Edgars SUNA, Robert STAVENGER, Mathias Winterhalter, and matthieu REFREGIERS
DOI: 10.26508/lisa.201800242

Corresponding author(s): Jean-Marie Pages, Faculté de Pharmacie

Review Timeline:

Submission Date:	2018-11-13
Editorial Decision:	2018-12-08
Revision Received:	2018-12-13
Editorial Decision:	2018-12-18
Revision Received:	2018-12-19
Accepted:	2018-12-19

Scientific Editor: Andrea Leibfried

Transaction Report:

December 8, 2018

Re: Life Science Alliance manuscript #LSA-2018-00242-T

Dr. Jean-Marie Pages
Faculté de Pharmacie
UMR_MD1, U-1261, Membranes and Therapeutic Targets
27 Bd Jean Moulin
Marseille 13005
France

Dear Dr. Pages,

Thank you for submitting your manuscript entitled "Mechanistic aspects of maltotriose-conjugate translocation to the Gram-negative bacteria cytoplasm" to Life Science Alliance. The manuscript was assessed by expert reviewers, whose comments are appended to this letter.

As you will see, the reviewers appreciate your work and provide constructive input on how to further strengthen your manuscript. A few clarifications and improvements are needed. I would thus like to invite you to provide a revised version of your manuscript, addressing the individual concerns and suggestions made by the reviewers.

Thank you for this interesting contribution to Life Science Alliance. We are looking forward to receiving your revised manuscript.

Sincerely,

- A letter addressing the reviewers' comments point by point.
- An editable version of the final text (.DOC or .DOCX) is needed for copyediting (no PDFs).
- High-resolution figure, supplementary figure and video files uploaded as individual files: See our detailed guidelines for preparing your production-ready images, <http://life-science-alliance.org/authorguide>
- Summary blurb (enter in submission system): A short text summarizing in a single sentence the study (max. 200 characters including spaces). This text is used in conjunction with the titles of papers, hence should be informative and complementary to the title and running title. It should describe the context and significance of the findings for a general readership; it should be written in the present tense and refer to the work in the third person. Author names should not be mentioned.

B. MANUSCRIPT ORGANIZATION AND FORMATTING:

Full guidelines are available on our Instructions for Authors page, <http://life-science-alliance.org/authorguide>

Reviewer #1 (Comments to the Authors (Required)):

In this manuscript, the authors explore alternative uptake pathways for future drug conjugate therapies using the E. coli maltose transport system and maltodextrin molecules coupled to a fluorophore. This study uses state of the art single cell fluorescence techniques to measure the accumulation of two maltodextrin conjugates (Cpd-1 and Cpd-2) within E. coli cells. Biochemical

techniques, electrophysiology and fluorescence microscopy both at the population and at the single cell level are used to gain information on the kinetics of accumulation and on the distribution of the two compounds in the three bacterial compartments. The authors show that the smaller Cpd-1 conjugate diffuses efficiently across the LamB porin and enables its own uptake by induction of the maltose transport system.

This study paves the way for future drug delivery therapies, as initiated with the siderophore-drug conjugates, but using the maltose transport system as the hijacked uptake route. This pathway has indeed been explored previously using a similar maltotriose molecule conjugated to trimethoprim as a cargo. This conjugate was shown to be taken up by *E. coli* and was tested in an *in vivo* model. The current manuscript goes deeper in the analysis of the uptake process by showing at the single cell level the transport of the compound into the periplasm and the cytoplasm and its impact on the regulation of the maltose transport system.

Based on data from Fig 2B it seems that induction of LamB alone provides an optimal fluorescence signal, which is not further increased by induction of MalE. Does this mean that MalE retains the compound preventing its transport into the cytosol ?

Fig 2 B and C: the amount of LamB seems to be higher in the wt strain induced by maltose than in the lamB mutant in which LamB is induced from the plasmid, although accumulation seems to be higher in the latter condition. Are these blots comparable? Where is the loading control, for instance OmpC or OmpF, as comparator as shown in Fig S1 ? If these are from the same samples, then they should be combined and shown in Fig. 2.

What is the fluorescence signal for the lamB mutant induced by maltose alone as shown in Figure S3? If there is none, as expected, it must be stated in the text.

Fig. 2B: How do the authors explain the increase in fluorescence in strain RAM2807 (ara 0.2%) that carries the empty vector?

Fig. EV1B and C: shouldn't the colour labelling be opposite if excess maltose represses CPD-1 uptake (green vs blue)? This is not in agreement with the figure legend

Fig. EV2A and B: what is OSPT-1235 and 1236 on the y-axis of the figures ? Probably Cpd-1 and 2 ? Please change.

Fig. EV2C and D vs E: the fluorescence values for the standard curves of Cpd-2 (D) in individual cells seem to be higher than for Cpd-1 (B) although the accumulation kinetics (E) show the opposite ? Any explanation ?

Fig S5A and B : the diagram in panel B shows linearity only up to 5 μ M although concentrations used in panel A go up to 55 μ M. Is the linearity conserved above 5 μ M ? Meaning that saturation shown in panel A could result from non-linearity in fluorescence at higher CPD-1 concentrations.

Fig. 6. The induction of MalE is interesting and important. What about the other components of the inner membrane transport complex. A qPCR for malF and malG should be performed

Was the resistance to hydrolysis (enzymatic or extreme pH conditions) of the ether bond tested in the linker of Cpd-1 and Cpd-2? Meaning could the fluorescence label be released in the periplasm and diffuse alone into the cytosol (its very hydrophobic) ?

Minor comments

P19, 1st paragraph: CpD-1 conc is probably 55 μ M and not millim ?

Reviewer #2 (Comments to the Authors (Required)):

This manuscript focuses on investigating the ability of maltotriose-erythrin and maltohexaose-erythrin to enter E.coli. Experiments are performed with LamB mutants, which demonstrate that the probes are taken up via the maltodextrin transporter. In addition, microscopy experiments are performed that investigate the intracellular location of the probes, and determine that the probes are entering the bacteria. A series of biophysical experiments were also performed to characterize the interaction of their probes with the LamB receptor. Finally, experiments are performed that investigate the relative kinetics of the probes, and suggest that maltotriose is the best maltodextrin targeting ligand.

Overall this is an excellent paper, which will be avidly read by people working in the area of bacterial targeting. The paper is fairly complete. However, some recent literature on maltodextrin targeting needs to be cited and discussed, in particular recent papers, such as J Nucl Med. 2017 Oct;58(10):1679-1684, and ChemMedChem. 2018 Feb 6;13(3):241-250, are relevant to this work and need to be discussed.

Re: Life Science Alliance manuscript #LSA-2018-00242-T
Reviewers' comments and corresponding responses indicated by >>

Reviewer #1 (Comments to the Authors (Required)):

In this manuscript, the authors explore alternative uptake pathways for future drug conjugate therapies using the *E. coli* maltose transport system and maltodextrin molecules coupled to a fluorophore. This study uses state of the art single cell fluorescence techniques to measure the accumulation of two maltodextrin conjugates (Cpd-1 and Cpd-2) within *E. coli* cells. Biochemical techniques, electrophysiology and fluorescence microscopy both at the population and at the single cell level are used to gain information on the kinetics of accumulation and on the distribution of the two compounds in the three bacterial compartments. The authors show that the smaller Cpd-1 conjugate diffuses efficiently across the LamB porin and enables its own uptake by induction of the maltose transport system. This study paves the way for future drug delivery therapies, as initiated with the siderophore-drug conjugates, but using the maltose transport system as the highjacked uptake route. This pathway has indeed been explored previously using a similar maltotriose molecule conjugated to trimethoprim as a cargo. This conjugate was shown to be taken up by *E. coli* and was tested in an *in vivo* model. The current manuscript goes deeper in the analysis of the uptake process by showing at the single cell level the transport of the compound into the periplasm and the cytoplasm and its impact on the regulation of the maltose transport system.

Based on data from Fig 2B it seems that induction of LamB alone provides an optimal fluorescence signal, which is not further increased by induction of MalE. Does this mean that MalE retains the compound preventing its transport into the cytosol ?

>> Our hypothesis is that in the presence of MalE/MalF/MalG, Cpd-1 is efficiently transported into cytoplasmic space where it could be metabolized by the degradative enzymes (see Fig 6 in the new version and the response to last point), this explains the reduced signal when the maltose operon is fully induced in addition to LamB.

Fig 2 B and C: the amount of LamB seems to be higher in the wt strain induced by maltose than in the lamB mutant in which LamB is induced from the plasmid, although accumulation seems to be higher in the latter condition. Are these blots comparable? Where is the loading control, for instance OmpC or OmpF, as comparator as shown in Fig S1 ? If these are from the same samples, then they should be combined and shown in Fig. 2.

>> It is difficult to systematically use OmpC and OmpF as comparator since when LamB is induced, *via* the maltose operon, we can observe a decrease of the OmpC-F porin expression (due to the protein balance in OM).

This is not a quantified result (blot) resulting from same samples, just a blot to check the OM protein expression. In order to not complicate the Fig.2, which is already complex, we prefer to maintain the selected format, and present independently Fig 2 and Fig S1.

What is the fluorescence signal for the lamB mutant induced by maltose alone as shown in Figure S3? If there is none, as expected, it must be stated in the text.

>> Only a very weak signal was obtained with the mutant, possibly due to non-specific adsorption (now mentioned in the text).

Fig. 2B: How do the authors explain the increase in fluorescence in strain RAM2807 (ara 0.2%) that carries the empty vector?

>> a possible explanation, the observed signal is a non-specific adsorption under these conditions despite the washing protocol.

Fig. EV1B and C: shouldn't the colour labelling be opposite if excess maltose represses CPD-1 uptake (green vs blue)? This is not in agreement with the figure legend

>> We agree, we have corrected this point in the new version. This figure EV1 corresponds to Fig S6 in the new version.

Fig. EV2A and B: what is OSPT-1235 and 1236 on the y-axis of the figures ? Probably Cpd-1 and 2 ? Please change.

>> It has been corrected in the new version. This figure EV2 corresponds to Fig 3 in the new version.

Fig. EV2C and D vs E: the fluorescence values for the standard curves of Cpd-2 (D) in individual cells seem to be higher than for Cpd-1 (B) although the accumulation kinetics (E) show the opposite ? Any explanation ?

>> For standard curves, the compounds (Cpd-1 and Cpd-2 in increasing concentrations) are added to bacterial lysates of different strains in order to obtain a robust standard assay. This is not an accumulation. The difference (C vs D) observed in the RFU is associated with the physicochemical properties of the assayed compounds. This figure EV2 corresponds to Fig 3 in the new version.

Fig S5A and B : the diagram in panel B shows linearity only up to 5 uM although concentrations used in panel A go up to 55 uM. Is the linearity conserved above 5 uM ? Meaning that saturation shown in panel A could result from non-linearity in fluorescence at higher CPD-1 concentrations.

>> The standard curve of figure S5B shows linearity up to 7.5 μ M, it must be noted that the concentrations used in panel A that go up to 55 μ M are external concentrations; the concentrations accumulated in cells are much lower.

Importantly, we have systematically checked that the fluorescence signal of the different samples measured in RFU fits in with in the linear part of the standard curve. If this was not the case, the samples were diluted until the fluorescence signal was in the linear part of the curve.

Fig. 6. The induction of MalE is interesting and important. What about the other components of the inner membrane transport complex. A qPCR for malF and malG should be performed.

>> I agree, but MalF and MalG are membrane proteins. In this case, as previously reported, transcriptomic studies generate only a partial view of protein expression since after translation they must be correctly inserted and assembled into the inner membrane (J Bacteriol. 2005, 187:2908-11). So, the appropriate assay is to check the presence of these proteins in the IM by using immunoblots, unfortunately we do not have anti-MalF or anti-MalG antiserum. This figure corresponds to Fig 7 in the new version.

Was the resistance to hydrolysis (enzymatic or extreme pH conditions) of the ether bond tested in the linker of Cpd-1 and Cpd-2? Meaning could the fluorescence label be released in the periplasm and diffuse alone into the cytosol (its very hydrophobic) ?

>> Not specifically, no. Note however the final step of synthesis of both compounds was room temp hydrolysis in strong base (LiOH) and no such cleavage was observed. Strongly acidic conditions (pH 1 or less, similar to stomach acid) might cleave the sugar from the rest of the molecule but is unlikely to cleave directly the ether link. However, extreme pH conditions are unlikely to be present in the bacterial systems report here, the *E. coli* periplasmic pH can be around 6.2 - 6.5 under the conditions used. We have not investigated enzymatic hydrolysis, but the ether bond in such linkers is not generally highly labile.

Regarding enzymatic hydrolysis, to our knowledge, etherases have been described in *E. coli*: these enzymes are involved in peptidoglycan recycling, especially during stationary growth phase, and they are located in cytoplasm (see *Microbiol Mol Biol Rev* 2008, 72:211–227). This enzyme location may also support our hypothesis regarding the Cpd-1 cytoplasmic degradation and we briefly mention this point in the discussion section of the new version.

Minor comments

P19, 1st paragraph: CpD-1 conc is probably 55 microM and not milliM ?

>> It has been corrected in the new version.

Reviewer #2 (Comments to the Authors (Required)):

This manuscript focuses on investigating the ability of maltotriose-erythrin and maltohexaose-erythrin to enter *E. coli*. Experiments are performed with LamB mutants, which demonstrate that the probes are taken up via the maltodextrin transporter. In addition, microscopy experiments are performed that investigate the intracellular location of the probes, and determine that the probes are entering the bacteria. A series of biophysical experiments were also performed to characterize the interaction of their probes with the LamB receptor. Finally, experiments are performed that investigate the relative kinetics of the probes, and suggest that maltotriose is the best maltodextrin targeting ligand.

Overall this is an excellent paper, which will be avidly read by people working in the area of bacterial targeting. The paper is fairly complete. However, some recent literature on maltodextrin targeting needs to be cited and discussed, in particular recent papers, such as *J Nucl Med.* 2017 Oct;58(10):1679-1684, and *ChemMedChem.* 2018 Feb 6;13(3):241-250, are relevant to this work and need to be discussed.

>> Thanks a lot for this information. These two papers described the use of labeled-maltodextrins for diagnosis and localization of bacterial infections in animal models (mouse). We have mentioned these papers in the Introduction section, but this is a little bit different to the aims of this work.

December 18, 2018

RE: Life Science Alliance Manuscript #LSA-2018-00242-TR

Dr. Jean-Marie Pages
Faculté de Pharmacie
UMR_MD1, U-1261, Membranes and Therapeutic Targets
27 Bd Jean Moulin
Marseille 13005
France

Dear Dr. Pages,

Thank you for submitting your revised manuscript entitled "Mechanistic aspects of maltotriose-conjugate translocation to the Gram-negative bacteria cytoplasm". As you will see, reviewer #1 appreciates the introduced changes, and we would be thus happy to publish your paper in Life Science Alliance pending final revisions necessary to meet our formatting guidelines:

Please add the information to your manuscript figure legends that Fig 2A RAM1292 LamB and MAIE blots have been reused in Fig S3.

A. FINAL FILES:

-- High-resolution figure, supplementary figure and video files uploaded as individual files: See our detailed guidelines for preparing your production-ready images, <http://life-science-alliance.org/authorguide>

B. MANUSCRIPT ORGANIZATION AND FORMATTING:

Full guidelines are available on our Instructions for Authors page, <http://life-science-alliance.org/authorguide>

Sincerely,

Andrea Leibfried, PhD
Executive Editor
Life Science Alliance
Meyerohofstr. 1
69117 Heidelberg, Germany
t +49 6221 8891 502
e a.leibfried@life-science-alliance.org
www.life-science-alliance.org

Reviewer #1 (Comments to the Authors (Required)):

The authors have responded to all the concerns of this reviewer. The authors made appropriate amendments in the text and added additional references that relate to maltodextrin transport.

December 19, 2018

RE: Life Science Alliance Manuscript #LSA-2018-00242-TRR

Dr. Jean-Marie Pages
Faculté de Pharmacie
UMR_MD1, U-1261, Membranes and Therapeutic Targets
27 Bd Jean Moulin
Marseille 13005
France

Dear Dr. Pages,

Thank you for submitting your Research Article entitled "Mechanistic aspects of maltotriose-conjugate translocation to the Gram-negative bacteria cytoplasm". It is a pleasure to let you know that your manuscript is now accepted for publication in Life Science Alliance. Congratulations on this interesting work.

DISTRIBUTION OF MATERIALS:

Again, congratulations on a very nice paper. I hope you found the review process to be constructive and are pleased with how the manuscript was handled editorially. We look forward to future exciting submissions from your lab.

Sincerely,

Andrea Leibfried, PhD
Executive Editor
Life Science Alliance
Meyerohofstr. 1
69117 Heidelberg, Germany
t +49 6221 8891 502
e a.leibfried@life-science-alliance.org
www.life-science-alliance.org